# Vicarious Offense and Noise Audit of Offensive Speech Classifiers: Unifying Human and Machine Disagreement on What is Offensive

**Tharindu Cyril Weerasooriya**♣* **Sujan Dutta**♣* **Tharindu Ranasinghe**◇
**Marcos Zampieri**♠ **Christopher M. Homan**♣ **Ashiqur R. KhudaBukhsh**♣
♣Rochester Institute of Technology
◇Aston University
♠George Mason University
cyril@mail.rit.edu, sd2516@rit.edu, t.ranasinghe@aston.ac.uk
mazgla@rit.edu, cmhvcs@rit.edu, axkvse@rit.edu

## Abstract

⚠This paper discusses and contains content that is offensive or disturbing. Offensive speech detection is a key component of content moderation. However, what is offensive can be highly subjective. This paper investigates how machine and human moderators disagree on what is offensive when it comes to real-world social web political discourse. We show that (1) there is extensive disagreement among the moderators (humans and machines); and (2) human and large-language-model classifiers are unable to predict how other human raters will respond, based on their political leanings. For (1), we conduct a *noise audit* at an unprecedented scale that combines both machine and human responses. For (2), we introduce a first-of-its-kind dataset[1] of *vicarious offense*. Our noise audit reveals that moderation outcomes vary wildly across different machine moderators. Our experiments with human moderators suggest that political leanings combined with sensitive issues affect both first-person and vicarious offense.

## 1 Introduction

Offensive speech on web platforms is a persistent problem with wide-ranging impacts (Benson, 1996; Chandrasekharan et al., 2017; Perez and Heater, 2021). Among many reasons why moderation fails to eliminate the problem, is the reality that people often disagree on what is offensive (Waseem, 2016; Ross et al., 2016). In this study, we break new ground by investigating disagreement through a large-scale *noise audit* and by introducing the notion of *vicarious offense*.

---

* Tharindu Cyril Weerasooriya and Sujan Dutta are equal-contribution first authors. Ashiqur R. KhudaBukhsh is the corresponding author.

[1]The dataset is available through https://github.com/Homan-Lab/voiced

### 1.1 Definitions

**Noise audit** While limited literature exists on investigating the generalizability of offensive speech detection systems across datasets (Arango et al., 2019), political discourse (Grimminger and Klinger, 2021; Maronikolakis et al., 2022), vulnerability to adversarial attacks (Gröndahl et al., 2018), unseen use cases (Sarkar and KhudaBukhsh, 2021), and geographic biases (Ghosh et al., 2021), to the best of our knowledge, no work exists on a comprehensive, in-the-wild evaluation of offensive speech filtering outcomes on large-scale, real-world political discussions. One key impediment to performing in-the-wild analysis of content moderation systems is a lack of ground truth. It is resource-intensive to annotate a representative dataset of social media posts to test content moderation at scale. Much to the spirit of the celebrated book, "Noise: A Flaw in Human Judgment" by Kahneman et al. (2021), we bypass this requirement through a *noise audit* of several well-known offensive speech classifiers on a massive social web dataset with documented political dissonance (KhudaBukhsh et al., 2021, 2022).

As defined in Kahneman et al. (2021), noise audit measures outcome variability across multiple (competent) decision systems. Kahneman et al. (2021) show, in real-world scenarios like judicial sentencing or insurance settlements that contain a diverse body of decisionmakers, that noise is widespread. Our paper seeks to understand how content moderation outcomes vary across different offensive speech classifiers (dubbed machine moderators) in political discourse at the granularity of individual social media posts.

**Vicarious offense** Existing literature reveals that annotator's gender, race, and political leanings may affect people's perception of what is offensive content (Cowan et al., 2002; Norton and Sommers, 2011; Carter and Murphy, 2015; Prabhakaran et al.,

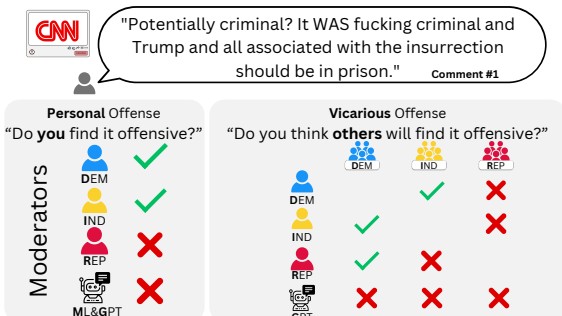

Figure 1: An illustrative example from YouTube comments on CNN news videos highlighting nuanced inconsistencies between machine moderators and human moderators with different political leanings. We use the majority vote to aggregate individual machine and human moderator's verdicts. The personal offense is listed in the left column outside the grid. The vicarious offense is presented in the right column with the grid. While Republicans feel that the comment *Potentially criminal? It WAS fucking criminal and Trump and all associated with the insurrection should be in prison.* will invite equal ire from the Independents, the Independents, actually, **do not** find it offensive. For the machine moderators (MM), we have nine MM $\mathcal{M}_j$ (Section 3.1) and ChatGPT API ($\mathcal{L}$), a large language model (LLM). Both types of MMs are able to detect personal offense; only the GPT model (Section 6) is able to identify the personal offense. The appendix contains more illustrative examples (Figure 17).

2021; Sap et al., 2022). We explore this phenomenon by introducing the notion of *vicarious offense* in which we ask a timely and important question: *how well do Democratic-leaning users perceive what content would be deemed as offensive by their Republican-leaning counterparts or vice-versa?*

For instance, Figure 1 shows the post, *Potentially criminal? It WAS fucking criminal and Trump and all associated with the insurrection should be in prison.*, and we imagine asking three human annotators—one Republican, one Democrat, and one independent—whether they think the post is personally offensive to them AND whether they think those with different political affiliations would find it offensive. Figure 1 shows that the Republican rater thinks it is personally offensive to them, and Independents would find it offensive, but not Democrats. We also ask a machine model whether the post is offensive (it thinks it is), and whether it is offensive to Republicans (yes), Democrats (no) and independents (no). We do NOT ask humans to rate how they expect machines would respond.

Documented evidence indicate that negative views towards the opposite political party have affected outcomes in settings as diverse as allocating scholarship funds (Iyengar and Westwood, 2015),

mate selection (Huber and Malhotra, 2017), and employment decisions (Gift and Gift, 2015). Is it possible that such negative views also affect our ability to perceive what we dub *vicarious offense*?

We thus break down our study on how political leanings affect perception of offense into two parts: (1) annotators answer questions relevant to direct, first-person perception of offense; (2) annotators answer about vicarious offense. For any given social media post $d$, we have two points of view. For each $\mathcal{X}, \mathcal{Y} \in \{$Republican, Democrat, Independent$\}$ where $\mathcal{X} \neq \mathcal{Y}$, we have: **Personal (or 1st person):** how offensive a(n) $\mathcal{X}$ finds $d$; and **Vicarious (or 2nd person):** how offensive a(n) $\mathcal{X}$ *thinks* a(n) $\mathcal{Y}$ would find $d$. With three distinct choices of $\mathcal{X}$ for the Personal, and six distinct choices for $(\mathcal{X}, \mathcal{Y})$ for the Vicarious, we have nine vantage points (see Figure 1).

Our study is the first of its kind to explore how well we can predict offense for others who do not share the same political beliefs. In the era of growing political polarization in the US where sectarian *us vs. them* often dominates the political discourse (Finkel et al., 2020), our study marks one of the early attempts to quantify how well the political opposites understand each other when asked to be on *their opposites'* shoes.

## 1.2 Research Questions and Contributions

**Research Questions** We address the following research questions:

- **RQ1:** *How aligned are offense predictions of machines moderators across different machine moderators?*

- **RQ2** *How aligned are offense predictions of human moderators across different political beliefs?*

- **RQ3** *How is the alignment between offense predictions of human moderators and machine moderators?*

**Contributions** Our contributions are as follows: (1) We conduct a *noise audit* at an unprecedented scale that reveals considerable variations in in-the-wild content moderation outcomes across nine different well-known machine moderators; (2) We conduct a detailed annotation study to understand the phenomenon of vicarious offense and release the first-of-its-kind dataset where annotators label both first-person and *vicarious* offense.

Unlike most studies on political biases which proffer a binarized world-view of US politics, we consider all three key players in the US democracy: the Democrats, the Republicans, and the Independents; (3) We release an annotator-level dataset, dubbed VOICED (**V**icarious **O**ffense **I**dentification **C**orpus for **E**stimating **D**isagreements), available at: `https://github.com/Homan-Lab/voiced`, consisting of 2,310 social web posts, that sheds critical insights into how well political groups understand each other's perception of offense in general and also when sensitive issues such as reproductive rights or gun control/rights are in the mix; and (4) Finally, we close the loop by analyzing how human moderators (dis)agree with their machine moderator counterparts on what is offensive, including the use of ChatGPT as a vicarious predictor.

## 2 Related Work

We outline some of the key inspirations of our work in this section. Throughout the paper, we cite several other relevant papers.

The primary inspiration of our *noise audit* approach is the celebrated book, "Noise: A Flaw in Human Judgment" by Kahneman et al. (2021). However, to our knowledge, our paper is the first to apply this method in auditing offensive speech classifiers bypassing the requirement of a large number of annotated samples.

Our research benefits from the vast literature on offensive speech classification that has yielded several valuable datasets that we use in this paper (Davidson et al., 2017; Mandl et al., 2020; Basile et al., 2019; Mathew et al., 2021; Zampieri et al., 2019; Kumar et al., 2020; Qian et al., 2019). However, while limited literature exists on investigating the generalizability of offensive speech detection systems across data-sets (Arango et al., 2019), vulnerability to adversarial attacks (Gröndahl et al., 2018), unseen use cases (Sarkar and KhudaBukhsh, 2021), and geographic biases (Ghosh et al., 2021), to our knowledge, no in-the-wild, comprehensive performance evaluation of offensive speech classifiers on political discourse has been conducted before.

Annotator subjectivity has been widely studied in (Prelec, 2004; Pavlick and Kwiatkowski, 2019; Dumitrache et al., 2018; Poesio et al., 2019; Liu et al., 2019; Weerasooriya et al., 2020; Basile, 2020; Weerasooriya et al., 2023). Specifically, how political influence may factor in offensive speech annota-

tion has been studied before (Sap et al., 2021). Our work contrasts with Sap et al. (2021), in the following key ways: (1) our introduction of *vicarious offense*, a novel offense perspective not considered heretofore; (2) the inclusion of the Independents in our annotation study; and (3) a unified analysis of alignment between machine and human moderators.

## 3 Methods

### 3.1 Machine Moderators (a.k.a., MMs)

We investigate nine open sourced offensive language identification models as machine moderators. From these models, we trained eight models on the following well-known offensive speech datasets obtained from Twitter, Facebook, Gab, Reddit, and YouTube: (1) AHSD (Davidson et al., 2017); (2) HASOC (Mandl et al., 2020); (3) HatEval (Basile et al., 2019); (4) HateXplain (Mathew et al., 2021); (5) OLID (Zampieri et al., 2019); (6) TRAC (Kumar et al., 2020); (7) OHS (Qian et al., 2019); (8) TCC.[2] Following work by Ranasinghe and Zampieri (2020), we transform the labels of instances present in above datasets into two broad labels: offensive and not offensive, which correspond to the level A of the popular OLID taxonomy (Zampieri et al., 2019) widely used in offensive speech classification. We trained BERT-LARGE-CASED models on each of the training sets in these datasets following a text classification objective. As the (9) and final model we used publicly available Detoxify (Hanu and Unitary team, 2020), which is a ROBERTA-BASE model trained on data from Unintended Bias in Toxicity Classification by Jigsaw Team (cjadams et al., 2017), it is a dataset that is used in training pipelines of Perspective API (see Figure 15 in Appendix for analysis using Perspective API).

### 3.2 Human Moderators

**Annotation Study Design.** Our survey is grounded in prior human annotation literature studying subjectivity in offense perception (Sap et al., 2020; Kumar et al., 2021).[3] We host the survey in Qualtrics and make it only visible to users registered as living in the US. We set restrictions on MTurk due to the nature of the study. We release

---

[2] Available at `https://www.kaggle.com/c/jigsaw-toxic-comment-classification-challenge`
[3] A version of this survey can be found on this link `https://github.com/Homan-Lab/voiced`

our study in batches of 30 data items in total with 10 items from each news outlet but with varying levels of MM disagreements. Each batch consisted of 10 instances each from $\mathcal{D}_{offensive}$, $\mathcal{D}_{debated}$, and $\mathcal{D}_{notOffensive}$. Each instance is designed to be annotated by 20 annotators. We not only asked if each item was offensive to the annotator, but how someone with a different political identity would find it offensive. Our study was reviewed by our Institutional Review Board and was deemed as exempt.

**Pilot Study and Annotator Demographics.** Since MTurk has documented liberal bias (Sap et al., 2022), we first conduct a pilot study with 270 examples (nine batches) to estimate political representation. 117 unique annotators participate in this pilot. We observe a strong Democratic bias in the annotator pool (66% *Dem*ocrat, 23% *Rep*ublican, and 11% *Ind*ependent).[4]

To ensure comparable political representation, we set restrictions for the subsequent annotation batches to have at least six annotators from each political identity (18 annotators in total). The remaining two spots are given to the annotators who first accept the jobs regardless of their political identity. We also re-run batches from our pilot study to ensure they all contain at least six annotators from each political identity.

**Final Study.** Adding the political identity-based restrictions aided in building a representative dataset for this work (see Appendix C). We conducted a total of 37 batches of our survey of 30 items each, following the same survey structure as the pilot study.

Demographics of the final study annotator pool;
• *Political Leaning*: 35% (267) registered as *Dem*ocrats, 35% (266) as *Rep*ublicans, and 30% (220) as an *Ind*ependent.
• *Gender*: 47% Female, 53% male, and one non-binary annotator.
• *Race*: Similar to the pilot study, majority of the annotators are White or Caucasian, with limited representations from the Asian, Black or African American, and American Indians communities (in line with Sap et al. (2020)).
• *Age*: The study had annotators from all age groups above 18 years, majority of the annotators were from the age group 25-34.

---

[4]Detailed annotator demographics of the pilot study and the final study are presented in the Appendix C.

**Annotator Compensation.** We compensate the annotator 0.1 USD for each instance. Each batch with 30 instances would thus fetch 3 USD. Compensation is grounded in prior literature and is discussed in detail in the Appendix. We allow the annotators to leave a comment on our study at the end. No annotator complained about compensation while many praised our task's novelty.

## 4 Data

### 4.1 Dataset for Noise Audit

We evaluate our moderators on a dataset of more than 92 million comments on 241,112 news videos hosted between 2014, January 1 to 2022, Aug 27 by the official YouTube channels of three prominent US cable news networks: CNN, Fox News, and MSNBC. We consider this dataset because of the broad participation, topical diversity, and recent literature indicating substantial partisan and ideological divergence in both content and audience in these news networks (Stanley, 2012; Bozell, 2004; Gil de Zúñiga et al., 2012; Hyun and Moon, 2016; Dutta et al., 2022; Ding et al., 2023), with prior work reporting considerable political dissonance (KhudaBukhsh et al., 2021, 2022). Overall, our dataset comprises three million randomly sampled comments, one million from each of the three YouTube channels denoted by $\mathcal{D}_{cnn}$, $\mathcal{D}_{fox}$, and $\mathcal{D}_{msnbc}$, respectively. Temporal and linguistic analyses on the dataset are included in the Appendix B.

### 4.2 Dataset for Human Moderators

In order to compare and contrast machine moderation and human moderation, we first construct a representative set of easy and challenging examples from the machine moderators' perspective. For each corpus $\mathcal{D}$, we conduct a stratified sampling from three subsets: (1) a subset where most MMs agree that the content is not offensive (denoted by $\mathcal{D}_{notOffensive}$); (2) a subset where most MMs agree the content is offensive (denoted by $\mathcal{D}_{offensive}$); and (3) a subset in the twilight zone where nearly half of the models agree that the content is offensive with the other half agreeing that it is not (denoted by $\mathcal{D}_{debated}$). Formally,

$$d \in \begin{cases} \mathcal{D}_{notOffensive} & \text{if } 0 \leq \mathit{offenseScore}(d) \leq 1, \\ \mathcal{D}_{debated} & \text{if } \left\lfloor \frac{N}{2} \right\rfloor \leq \mathit{offenseScore}(d) \leq \left\lceil \frac{N}{2} \right\rceil, \\ \mathcal{D}_{offensive} & \text{if } N-1 \leq \mathit{offenseScore}(d) \leq N, \end{cases}$$

where $N$ denotes the total number of offensive speech classifiers considered (in our case, $N = 9$),

and *offenseScore*($d$) returns the number of MMs that deem $d$ offensive ($[0, N]$).

We have three news outlets, three sub-corpora defined based on MM disagreement, and five time periods yielding 45 different combinations of news networks, temporal bins, and MM disagreement. We weigh each of these combinations equally and sample 1,110 comments ($\mathcal{D}_{general}$). In addition, we sample 600 comments with the keyword gun ($\mathcal{D}_{gun}$) and 600 more with the keyword abortion ($\mathcal{D}_{abortion}$) to shed light on human-machine disagreement on hot-button issues. Filtering relevant content by a single, general keyword has been previously used in computational social science literature ([Halterman et al., 2021](); [Dutta et al., 2022]()). It is a high-recall approach to obtain discussions relevant to reproductive rights and gun control/rights without biasing the selection towards event-specific keywords (e.g., Row v. Wade or Uvalde).

## 5 Results

### 5.1 Machine Moderators in the Wild

**RQ1:** *How aligned are offense predictions of machines moderators across different machine moderators?*

Figure [2]() summarizes the pairwise agreement results (Cohen's $\kappa$) on the entire dataset. Results restricted to a specific news network are qualitatively similar. Our results indicate that no machine moderator pair exhibits substantial agreement ($\kappa$ score $\geq 0.81$), only a handful exhibit moderate agreement ($0.41 \leq \kappa$ score $\leq 0.60$), and several pairs exhibit fair, slight, or no agreement. When we quantify agreement across all machine moderators, the overall Fleiss' $\kappa$ across $\mathcal{D}_{cnn}$, $\mathcal{D}_{fox}$, and $\mathcal{D}_{msnbc}$ are 0.27, 0.25, and 0.22, respectively.

We next examine the distribution of machine moderators' aggregate verdicts on individual comments. As already mentioned, *offenseScore*($d$) returns the number of MMs that deem $d$ offensive ($[0, N]$).

A large fraction (nearly half) of the content is not flagged by any of the MMs, whereas a minuscule proportion (0.03%) is flagged as offensive by all. The content with *offenseScore* = 1 ($\approx 17.5\%$) is particularly interesting. It indicates only one of the nine MMs marks these comments as offensive. Therefore, the moderation fate of every comment in this bin is highly volatile. If any other MM than the one that flags it is deployed, the comment will not be censored. We also observe that 10.1% of the

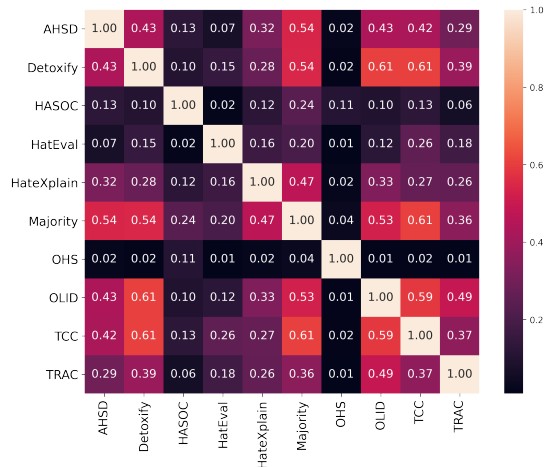

Figure 2: Agreement between machine moderators. A cell $\langle i, j \rangle$ presents the Cohen's $\kappa$ agreement between machine moderators $\mathcal{M}_i$ and $\mathcal{M}_j$. Majority is a machine moderator that takes the majority vote of the nine individual machine moderators.

content has *offenseScore* $\in \{4, 5\}$. These are the comments on which the MMs have maximal disagreement. To summarize, a large fraction of the social web represents disputed moderation zone and possibly requires human moderators' assistance. In what follows, we investigate how human moderators fare when they are tasked with the difficult job of determining offense in political discourse.

### 5.2 How Well Do Human Moderators Agree?

**RQ2:** *How aligned are offense predictions of human moderators across different political beliefs?*

We break **RQ2** down into two sub-parts – the first (**RQ2a**) investigating first-person offense, the second one (**RQ2b**) examining vicarious offense.

**RQ2a:** *How aligned are different political identities on the perception of offense in first-person?* Figure [3]() (in Appendix Table [7]()) summarizes the confusion matrices between human annotators of different political identities. We first observe that for any pair of political identities, the human-human agreement is higher than the best human-machine agreement achieved in our experiment. We next note that while human-human agreement is generally higher than human-machine agreement, the highest human-human Cohen's $\kappa$ achieved between the Independents and Democrats (0.43) is still at the lower end and is considered as moderate agreement ([McHugh, 2012]()). Within the political identity pairs, the Democrats and the Independents are most aligned on their perception of offense. This result is not surprising. Historically, Independents lean more toward the Democrats than the Republicans

as evidenced by the Gallup survey where $47.7 \pm 2.98\%$ of the Independents report that they lean Democrat as opposed to $42.3 \pm 3.08\%$ Independents reporting leaning Republican (Gallup, 2022).

**RQ2b:** *Which political identity best predicts vicarious offense?* In our vicarious offense study, we request the annotators to predict out-group offense labels. Hence, we have information about say, what Democrats believe Republicans find as offensive. Since we also have first-person perspectives from the Republicans, we can tally this information with the first-person perspective to find out how well annotators understand the *political others*.

We indicate the vicarious offense predictor in superscripts for clarity. Republicans$^{Dem}$ means Democrats are predicting what Republicans would find offensive. Figure 4 and Table 8 (in Appendix) indicates that Republicans are the worst predictors of vicarious offense. On both cases of predicting vicarious offense for the Democrats and the Independents, they do worse than the Independents and the Democrats, respectively. We further note that while Independents and Democrats can predict vicarious offense for each other reasonably well, they fare poorly in predicting what Republicans would find offensive. Hence, in terms of vicarious offense, Republicans are the least understood political group while they also struggle the most to understand others.

Finally, we present a compelling result that shows why inter-rater agreement could be misleading. Figure 4 and Table 9 (Appendix) suggest that the Democrats and Independents have the highest agreement on what Republicans would find as offensive. However, Table 8 already shows that neither the Democrats nor the Independents understand well what Republicans actually find offensive. Hence, if we have a pool of web moderators comprising only Democrats and Independents, their evaluations of what Republicans find as offensive will be reasonably consistent; however, it may not reflect what Republicans truly find as offensive.

## 5.3 Machine and Human Moderators

**RQ3:** *How is the alignment between offense predictions of human moderators and machine moderators?* Recent reports indicate increasing scrutiny on big-tech platforms' content moderation policies (Douek, 2021). The discussions center around two diametrically opposite positions: these platforms are not doing enough, or they need to do

| Machine moderators: *notOffensive* \| Human Moderators: *offensive* |
|---|
| So a woman wants an abortion, and SHE happens to have an airplane ticket that SHE got. The airline company can be sued? |
| Maybe read the U.S. Constitution. Abortion is not in it, unlike the 2nd Amendment. Please stop showing planned parenthood ads. I don't agree with abortion. |
| republicans are so far right that facts don't matter anymore. it is political theater. 0 dollars of Federal money has been spent on abortion. the war is on abortions. other people trying to dictate how other people should live their lives. |

Table 1: Illustrative examples highlighting disagreement between machine moderators and human moderators on $\mathcal{D}_{abortion}$. The blue, yellow, and the red cells consider Democrats, Independents, and Republicans human moderators, respectively

more. On one hand, certain camps believe that web platforms censor a particular political believers unjustly more (Barrett, 2022). On the other hand, different groups often believe that poor platform censorship led to some of the recent political crises (Derysh, 2021). Figure 3 (in Appendix Table 6) examines to which political party machine moderators are most aligned with. We observe that while all three political identities have comparable agreement with machines on what is offensive, Republicans align slightly more with the machines on what is not offensive. Existing literature hypothesized that conservatives may focus more on linguistic purity than the liberals while determining toxic content (Sap et al., 2022). We note that of all the instances in $\mathcal{D}_{general}$ that contains the keyword fuck, 94% of them were marked as offensive by the Republicans whereas Democrats marked 88% of them as offensive.

Consider the following examples without any profanity; yet the Democrats marked them as *offensive* but the Republicans did not:
• *More fear-mongering. only .oo6% of people die of covid. Russia has no reason to lie about the death total.*
• *Diversity====Less White people, White shaming. I see everyone as Americans not by their Skin color, The real Racist say black, white, brown pride.* It is possible that the Democrats found these examples offensive because they did not align with their liberal views.

We notice that some obfuscated profanity escaped machine moderators while human moderator groups caught them (e.g., cocksuxxxer, or 3-letter company). We also observe that hu-

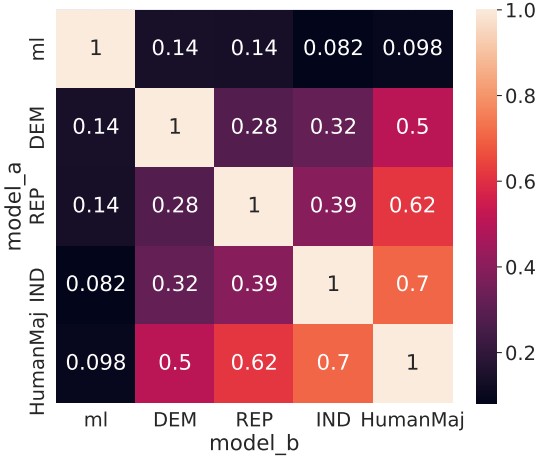

Figure 3: Agreement between machine moderators and human moderators. A cell $\langle i, j \rangle$ presents the Cohen's $\kappa$ agreement between moderators. ML is a machine moderator that takes the majority vote of the nine individual machine moderators. The majority responses from the human moderators are included in the figure Democrat (DEM), Republican (REP), Independent (IND), and Human Majority (HumanMaj).

mans having deeper contexts allows them to respond differently. A dismissive comment about Caitlyn Jenner, a transgender Olympic Gold Medalist, is unanimously marked by all groups of human moderators as *offensive* which the machine moderators marked as *notOffensive* (see Table 1, Table 2, and see Figure 17).

## 5.4 On Censorship and Trust in Democracy

Beyond first-person and vicarious offense, we ask the annotators a range of questions on censorship and election fairness. We explicitly ask every annotator if she believes the comment should be allowed on social media. We find that of the comments individual political groups marked as offensive on $\mathcal{D}_{general}$, the Democrats, Republicans, and Independents believe 23%, 23%, and 17% , respectively, should not be allowed on social media. This implies that in general political discussions, Independents are more tolerant than the two political extremes on what should be allowed on social media. On $\mathcal{D}_{gun}$, the Republicans exhibit slightly more intolerance than Democrats and Independents and want to remove 26% of the offensive content as opposed to 23% and 14% by the Democrats, and the Independents, respectively. However, on $\mathcal{D}_{abortion}$ the Democrats exhibit more intolerance seeking to remove 23% of the offensive content as opposed to 21% from both Independents and Republicans. We note that Independents are much more sensitive to reproductive rights than gun control/rights or general political discussions. Our study thus sug-

gests that content moderation is a highly nuanced topic where different political groups can exhibit different levels of tolerance to offensive content depending on the issue.

*What is offensive* and *should this offensive post be removed from social media* can be subjective, as our study indicates. However, when we ask the annotators about fairness of the 2016 and 2020 elections, we notice a more worrisome issue: eroding trust in democracy. 5% and 10% of the annotators believe that the 2016 and 2020 elections were not conducted in a fair and democratic manner, respectively. Democrats doubt the fairness of 2016 election more while the Republicans doubt the fairness of 2020 election. This result sums up the deep, divergent political divide between the left and the right in the US and asks all the stakeholders – social media platforms, social web users, media, academic and industry researchers, and of course the politicians – to think about how to improve political discourse and restore trust in democracy.

## 5.5 Ablation Studies

When we consider sensitive issues, we find the overall disagreement observed in our general dataset worsens further. Table 5 contrasts the pairwise disagreement between human-human moderators and human-machine moderators across $\mathcal{D}_{general}$, $\mathcal{D}_{abort}$, and $\mathcal{D}_{gun}$. We first observe that machine-human agreement is substantially lower on the issue-specific corpora across all political identities. We next note that some of the moderator pairs achieved negative Cohen's $\kappa$ values on $\mathcal{D}_{gun}$ even on first-person offense perspective indicating considerable disagreement (McHugh, 2012).

The pairwise group dynamics behave differently on different issues. While Independents exhibit considerable agreement with Republicans on $\mathcal{D}_{abortion}$, they show little or no agreement on $\mathcal{D}_{gun}$. Interestingly, while neither the Republicans nor the Independents agree a lot with a Democrats on $\mathcal{D}_{abortion}$ these two groups (Independents and Republicans) are well-aligned on what Democrats would find offensive in $\mathcal{D}_{abortion}$. However, once again, when we tally that with what the Democrats actually find as offensive, we see the agreement on the pairs $\langle$Democrats, Democrats$^{Rep}\rangle$ and $\langle$Democrats, Democrats$^{Ind}\rangle$ are substantially lower.

Table 1 lists few instances that machine moderators marked as *notOffensive* however, human moderators belonging to specific political identities

marked them as offensive.

We conduct noise audits on a broad range of datasets (Twitter, GAB, Reddit, YouTube) and observe qualitatively similar results (presented in the Appendix B.3).

# 6 Discussion and Conclusion

In this paper, we present two novel perspectives on moderating social media political discourse: disagreement between machine moderators, and disagreement between human moderators. Our key contributions are (1) a comprehensive noise audit of machine moderators; (2) VOICED, an offensive speech dataset with transparent annotator details; (3) a novel framework of vicarious offense; and (4) a focused analysis of moderation challenges present in dealing with sensitive social issues such as reproductive rights and gun control/rights.

Traditional supervised learning paradigm assumes existence of *gold standard* labels. While recent lines of work have investigated disagreement among annotators that stems from the inherent subjectivity of the task (Pavlick and Kwiatkowski, 2019; Ghosh et al., 2021; Prabhakaran et al., 2021; Davani et al., 2022), our analyses of political discussions on highly sensitive issues reveal that there could be practically no agreement among annotator groups and depending on who we ask, we can end up with wildly different *gold standard* labels reminiscent of *alternative facts* (Jaffe, 2017). Our current work is primarily descriptive. We address the elephant in the room and quantify the challenges of offensive content moderation in political discourse both from the machine and human moderators' perspectives. We believe our dataset will open the gates for modeling ideas considering multiple vantage points and yielding more robust systems.

Our study raises the following points to ponder upon.

• *Issue focused analysis:* In Section 5.5, our study barely scratches the surface of issue-focused analysis. Studies show that there are political disagreement between the left and the right on several other policy issues that include immigration (Card et al., 2022), climate change (Fisher et al., 2013), racism in policing (Dutta et al., 2022), to name a few. We believe our study will open the gates for follow on research expanding to more issues.

• *Style vs content:* We observed an important interplay between the style and the content of posts

| Machine moderators: *offensive* \| Human Moderators: *notOffensive* |
|---|
| Republicunts n Evangelicunts are a scourge to JesusBabies snatched from the parents is worse then abortion u shameless bastards |
| President Pussygrabber is a Star. We can allow him to murder a woman having an abortion on 5th Avenue and |
| Dickbag trump is horrible for our country. He is a lying con man who have Americas dumbest supporters |
| its ALL Bidens fault with his open borders!!! TRUMP WAS WAY BETTER THAN THIS IDIOT |
| YOU STUPID DEMS NEED TO LOOK AT WHATS HAPPENING TO EUROPE |
| This really stinks. The democraps really schitf their pants now!!! |

Table 2: Illustrative examples highlighting disagreement between machine moderators and human moderators on $\mathcal{D}_{general}$. The blue and red cells consider Democrats and Republicans human moderators, respectively

particularly when it comes to polarizing topics and political preference. As evidenced in Table 2, our analysis reveals that the topic and targets of a potentially offensive post (e.g. a politician, a political party, etc.) seem to be more important to human moderators than to machine moderators as automatic methods often rely on the presence of profanity to assign a post offensive. This observation is in line with datasets and annotation taxonomies that consider the target of offensive posts as the central aspect of offensive speech identification such as OLID (Zampieri et al., 2019), HASOC (Mandl et al., 2020) and others. The new vicarious offense dataset presented in this paper is a valuable resource that can be used for further analysis.

• *Evaluating Vicarious Offense as a Perspective:* The vicarious offense is a novel perspective that is not captured by existing classification models used in our noise audit. In this section, we use ChatGPT API ($\mathcal{L}$)[5] (gpt3.5turbo,v 3.5), a large language model (LLM) to predict the vicarious offense that humans are able to annotate as a comparison. For a given comment ⟨comment⟩, we use the following prompt: "You are an emotionally intelligent assistant who is a registered partyA. Classify the intent of the user's text to be offensive/disrespectful to someone registered as partyB with ONLY ONE OF THE FOLLOWING CHOICES: 1. Very offensive, 2. Extremely offensive, 3. Slightly offensive, 4. Moderately offensive, or 5. Not at all offensive.

---

[5] https://platform.openai.com/docs/models/gpt-3-5

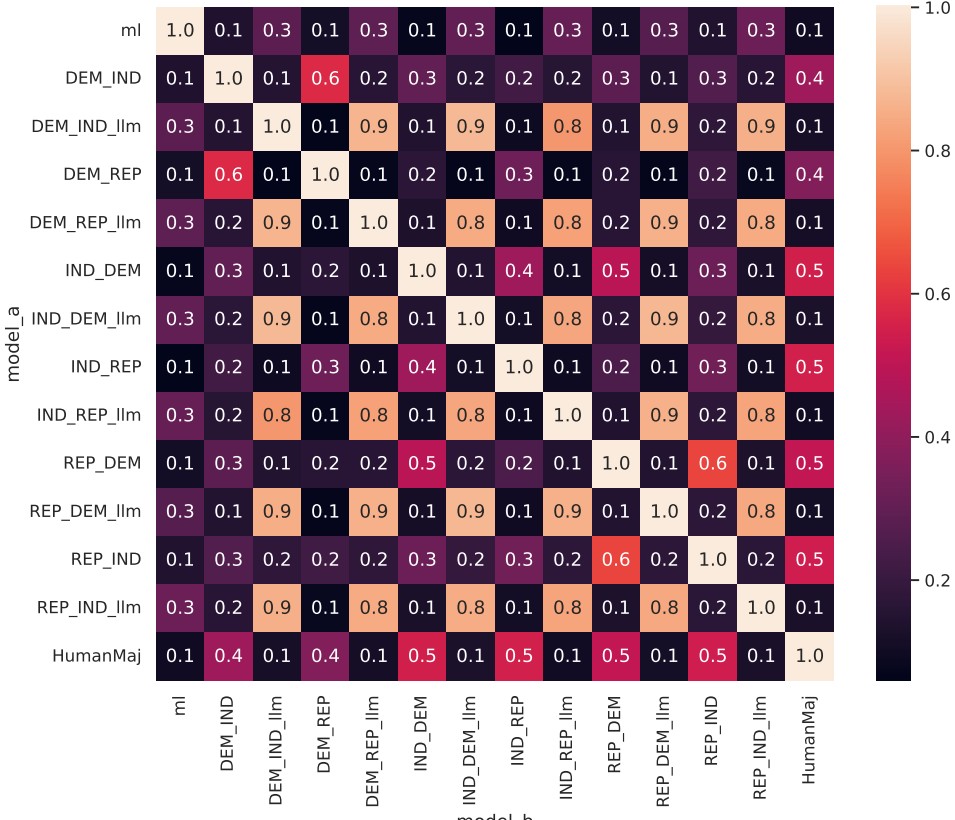

Figure 4: Agreement between human moderators and GPT model on vicarious offense. A cell $\langle i, j \rangle$ presents the Cohen's $\kappa$ agreement between LLM model $\mathcal{L}$ and $\mathcal{M}_j$. Here ml is a machine moderator that takes the majority vote of the nine individual machine moderators. The vicarious offense is denoted with partyA_partyB. The scores without a suffix after the vicarious offense denote the majority of the labels from the human moderators. The suffix llm denotes the vicarious offense predicted by the LLM. HumanMaj indicates the majority response considering all human moderators.

⟨comment⟩". In this prompt, partyA and partyB refer to a Democrat, a Republican, or an independent. Note that, we asked a similar question to the human moderators "How offensive do you think partyA will find this comment?". Results of this study are included in Figure 4.

In this case, the agreement of the $\mathcal{L}$ predictions of the vicarious offense was similar to what was observed in the human and ML moderators' agreement scores for the first-person offense identification. The LLM agreed less with the human moderators and more with the machine moderator majority. This adds evidence of how challenging offensive language identification is for machine moderators LLMs and non-LLMs alike for this dataset.

## Ethics Statement

Our study was reviewed by our Institutional Review Board and was deemed exempt. Our YouTube data is collected using the publicly available YouTube API. We do not collect or reveal any identifiable information about the annotators. Content moder-

ation can be potentially gruesome and affect the mental health of the moderators (Solon, 2017). We maintain a small batch size (30 YouTube comments), one-third of which is marked as *notOffensive* by almost all machine moderators to minimize the stress on annotators. In fact, many of the annotators left a comment at the end of the study indicating that they enjoyed this task. While our goal is to broaden our understanding of first-person and vicarious offense perception and has the potential to robustify machine moderators, any content filtering system can be tweaked for malicious purposes. For instance, an inverse filter can be made that filters out *notOffensive* posts while filtering in the *offensive* ones.

## Limitations

Our paper has the following limitations.
- **Fine-grained political identities:** Unlike recent papers studying political polarization that only consider conservative and liberal ideologies (Demszky et al., 2019), our paper marks one of the earliest at-

tempts to include the Independents in the mix. That said, political identities can be far more nuanced than the three options a voter can register for. For example, a human moderator can be fiscally conservative but socially liberal. Understanding both first-person and vicarious perspectives of offense considering more fine-grained political identities merits deeper exploration.

Our dataset was limited to English posts, primarily focusing on one platform (YouTube). Ideally, a study would consider other platforms and languages to provide a greater degree of external validity.

• *Beyond US politics and political polarization:* Finally, the framework of vicarious offense has a broader appeal. While we apply this to US politics, there is rising political polarization in many other countries (Gruzd and Roy, 2014; Silva, 2018). It does not also have to be always political differences. The vicarious offense framework can also be used to understand religious polarization (Palakodety* et al., 2020; Chandra et al., 2021; Saha et al., 2021).

• *Beyond US annotator populations:* We primarily conduct our human annotation studies through Amazon Mechanical Turk. Even though a broad annotator demographic was targeted in this study, an extension to this could be looking at the broader human annotation platforms while expanding it to other regions.

• *Learning to Predict Vicarious Offense:* The vicarious offense perspectives collected from the human annotators are challenging to teach an ML model. Our initial round of experiments included a closed LLM, ChatGPT as a baseline. ChatGPT is a black box in this setting, and we hope to explore this phenomenon further on other publicly available systems built on work by Ouyang et al. (2022) (InstructGPT) and Touvron et al. (2023) (Llama 2). Perspective API is another service that is used for predicting personal offense, we use `Detoxify` (Hanu and Unitary team, 2020) as a model that is trained on a dataset part of the Perspective API for consistent reproducibility. We include responses from ChatGPT and Perspective API for the human annotated VOICED dataset in Appendix Figure 15.

• *Modeling uncertainty* Vicarious offensive is related to concepts such as *soft labeling* (Vyas et al., 2020) and Bayesian truth serum (Prelec, 2004). We generally work with hard labels and where Bayesian truth serum is about predicting uncer-

tainty, we do not model uncertainty in this study.

## Acknowledgments

This research has been partially supported by an ESL Global Cybersecurity Institution Seed Fund and a gift from Google LLC. We would like to express our gratitude to the human annotators who took part in the study, as well as the anonymous reviewers for their valuable feedback and suggestions. Additionally, we thank Zohair Hassan and Sheeraja Rajakrishnan for their insightful discussions.

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

## A   Supplemental Material

### A.1   Experimental Setup and Reproducibility

The paper presents novel concepts that are more involved in understanding human annotations, but a significant portion of the experiments were conducted through ML models. The code to reproduce the first stage of the paper (noise audit) involves re-running the nine MM moderators through the YouTube dataset. The adaptation of the MMs for running the experiments is included in `https://github.com/Homan-Lab/voiced`. The run time for the predictions (1 million comments) on an Ubuntu 18.04, Intel i6-7600k (4 cores) at 4.20GHz, 32GB RAM, and nVidia GeForce RTX 2070 Super 8GB VRAM machine averaged 2 hours. The experimental code for analyzing the human responses is included in the repo.

## B   Machine Moderators

| Parameter | Value |
|---|---|
| adam epsilon | 1e-8 |
| batch size | 64 |
| epochs | 3 |
| learning rate | 1e-5 |
| warmup ratio | 0.1 |
| warmup steps | 0 |
| max grad norm | 1.0 |
| max seq. length | 256 |
| gradient accumulation steps | 1 |

Table 3: BERT Parameter Specifications.

They trained BERT on the datasets in Table 4, which has achieved state-of-the-art on a variety of offensive language identification tasks. From an input sentence, BERT computes a feature vector $h \in \mathbb{R}^d$, upon which we build a classifier for the task. For this task, we implemented a softmax layer, i.e., the predicted probabilities are $y^{(B)} = softmax(Wh)$, where $W \in \mathbb{R}^{k \times d}$ is the softmax weight matrix and $k$ is the number of labels. For the experiments, they used the BERT-LARGE-CASED model available in HuggingFace (Wolf et al., 2020).

To train the models they used a GeForce RTX 3090 GPU to train the models. They divided the dataset into a training set and a validation set using 0.8:0.2 split. For BERT they also used the same set of configurations mentioned in Table 3 in all the experiments. They performed *early stopping* if the validation loss did not improve over ten evaluation steps. All the experiments were conducted three

times, and the mean value is taken as the final reported result.

### B.1   Language Analysis on the Dataset of Human Moderators

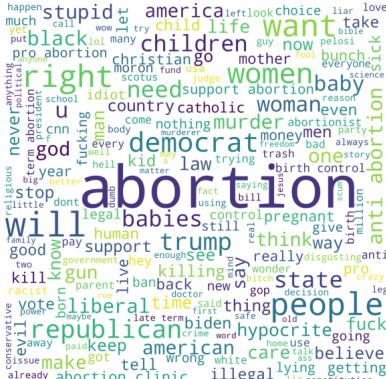

Figure 5: Word cloud on the Abortion dataset

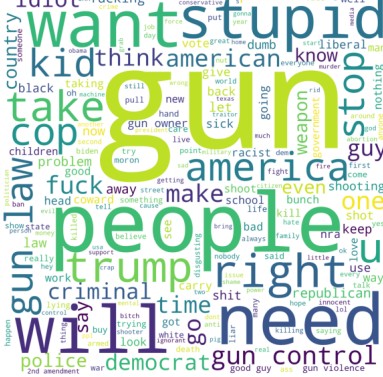

Figure 6: Word cloud on the Gun dataset

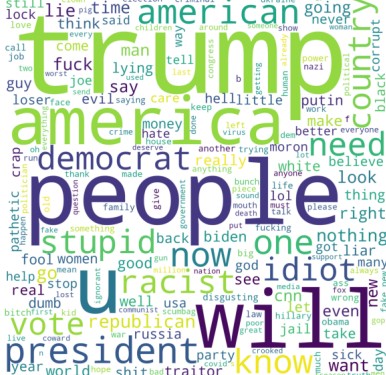

Figure 7: Word cloud on the General dataset

### B.2   Temporal Evolution of the Dataset

Figure 8 demonstrates that our dataset has sparse user engagement via comments between 2014 and 2018. Figure 9 demonstrates how machine moderators agree on the level of toxicity.

| Dataset | Training | | Testing | | Data Sources | Reference |
|---|---|---|---|---|---|---|
| | Inst. | OFF % | Inst. | OFF % | | |
| AHSD | 19,822 | 0.83 | 4,956 | 0.82 | Twitter | Davidson et al. (2017) |
| HASOC | 5,604 | 0.36 | 1,401 | 0.35 | Twitter, Facebook | Mandl et al. (2020) |
| HatEval | 9,000 | 0.42 | 1,434 | 0.42 | Twitter | Basile et al. (2019) |
| HateXplain | 11,535 | 0.59 | 3,844 | 0.58 | Twitter, Gab | Mathew et al. (2021) |
| OHS | 8,285 | 0.21 | 2,090 | 0.20 | Reddit | Qian et al. (2019) |
| OLID | 13,240 | 0.33 | 860 | 0.27 | Twitter | Zampieri et al. (2019) |
| TCC | 12,000 | 0.09 | 2,500 | 0.10 | Wikipedia Talk | URL[6] |
| TRAC | 4,263 | 0.20 | 1,200 | 0.42 | Facebook, Twitter, YouTube | Kumar et al. (2020) |

Table 4: The eight datasets with the number of instances (Inst.) in the training and testing sets, the OFF % in each set, the data source, and the reference.

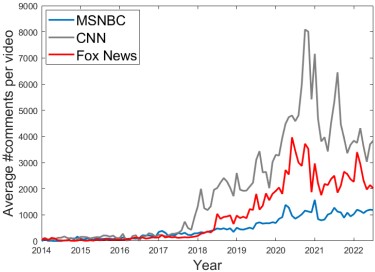

Figure 8: Temporal trend showing number of comments made about news videos on three news networks' official YouTube channels over time.

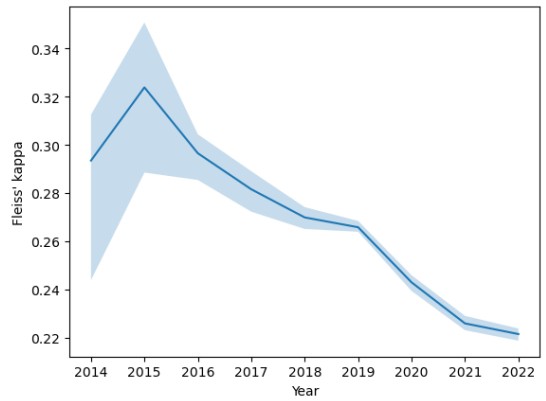

Figure 9: Temporal trend in agreement among the nine machine moderators on toxicity. The bands represent 95% confidence interval.

## B.3 Analyses on Other Datasets - MM Moderators

We conduct a large-scale analysis of YouTube channels of prominent US cable news networks. While these networks attract a diverse audience, a legitimate curiosity is whether the qualitative findings in our noise audit hold across other YouTube channels or platforms. With similar experimental settings, we conduct noise audits on the following nine datasets (Fleiss' $\kappa$ presented within parentheses): (1) YouTube comments on The Hill (0.18±0.002); (2) YouTube comments on political influencer Ben Shapiro (0.22±0.002); (3) YouTube comments on political influencer Bryan Tyler Cohen (0.26±0.002); (4) YouTube comments on BBC news (0.23±0.001); (5) YouTube comments on The Young Turks (0.26±0.001); (6) Newsmax TV (0.20±0.001); (7) a Twitter dataset on US politics (Chen et al., 2021) (0.25±0.002); (8) A Reddit r/politics dataset [7] (0.25±0.011); and (9) a Gab dataset (Zannettou et al., 2018) (0.29±0.001). Hence, our qualitative noise audit claims hold over several datasets across different social networks.

---

[7] https://figshare.com/articles/dataset/r_Politics_Dataset/20222433

## C   Human Moderators

This section includes in-depth analysis of the population of human annotators who participated in our study.

### C.1   Human Annotation Study

We conducted our human annotation study using Qualtrics for the surveys and Amazon Mechanical Turk for annotator recruitment. We have included a copy of the survey (in human readable format) as well as in a format that can be easily imported to Qualtrics in our code repo. The repo also includes resources on running a similar human annotation study.

### C.2   Distribution of Independents in Election Voting

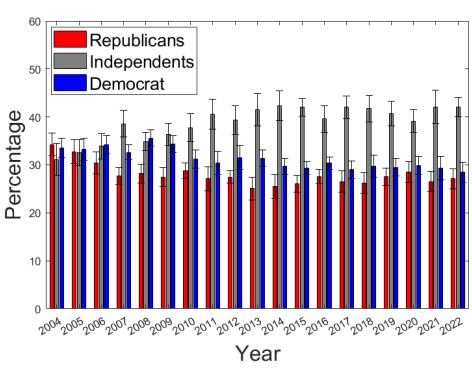

Figure 10: Distribution of political identities as reported in historical Gallup surveys (Gallup, 2022) over the last 19 years.

### C.3   Annotator Compensation

Initial pilot by the authors estimated 12$/hour compensation (completion time: 15 minutes; compensation: 3$/task) – more than US minimum wage (7.25$) and falls within the range reported in the literature (e.g., 6$/hour in Leonardelli et al. (2021); 7.25$/hour in Bugert et al. (2020); and 13$/hour in Bai et al. (2021)). The final study yielded 7.8$/hour median compensation (completion time: 23 minutes).

### C.4   Demographic Analysis

The distributions for gender did not show any imbalances as the population had equal representation from each gender (50% from each). Interestingly, the majority of the Democrat population was female and there was a higher population of male annotators who were Independents.

In Figures 12, we see the distributions for ages and race of the annotator population. The majority of the annotators belong to the white or Caucasian race and in the age group of 25-34. The overall annotator demographics are described in Figure 13.

We also asked the question if the presidential election in the years 2016 and 2020 was conducted in a fair and democratic manner. The interesting insight from these two figures show how the members of losing party thinks the election wasn't conducted in a fair manner. In 2016, the Republican president Donald J. Trump won the election and out of the annotators who thought it wasn't fair are Democrats, similarly in 2020 the Democratic nominee Joseph R. Biden Jr. won the election and majority of the annotators who believe election wasn't fair are Republicans.

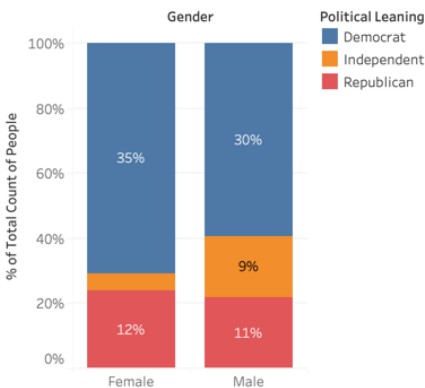

Figure 11: Distribution of the annotators based on their political leaning and gender. The % denotes total percentage from the whole population who belong to each subgroup.

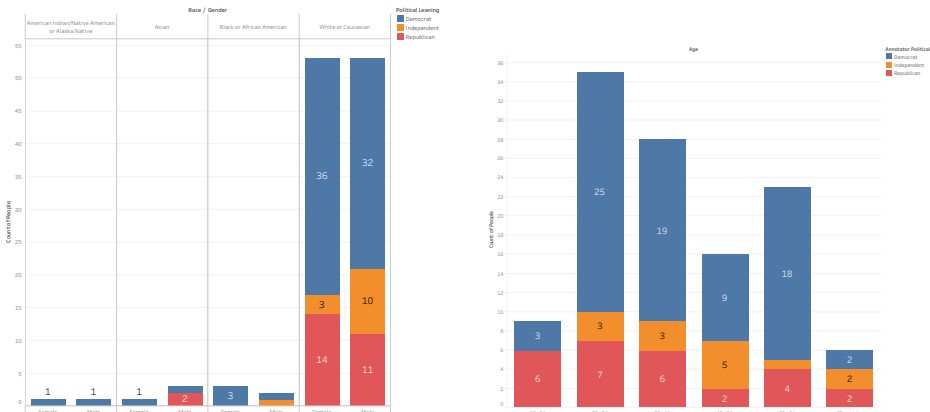

Figure 12: Distributions based on Race (left) and Age (right). The counts denotes number of annotators from the whole population who belong to each subgroup. The colors denote the political leaning.

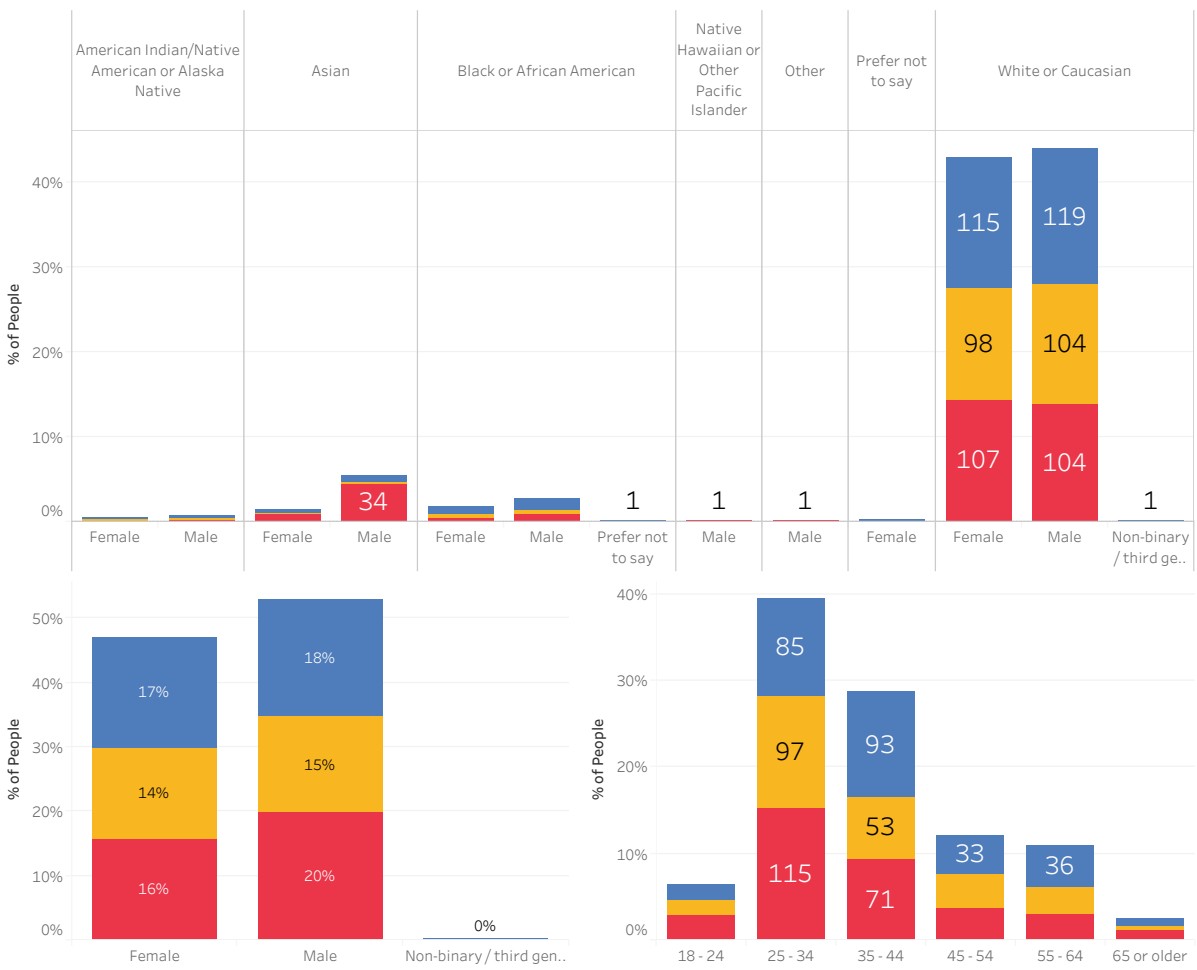

Figure 13: Overall demographic distribution of the annotator pool. The colors denote the political identities.

# D Annotator Agreement

## D.1 Disagreement Across Datasets - Raw Results

| Moderators | Moderators | $\mathcal{D}_{general}$ | $\mathcal{D}_{abortion}$ | $\mathcal{D}_{gun}$ |
|---|---|---|---|---|
| Machines | Republicans | 0.23 | 0.04 | 0.07 |
| Machines | Democrats | 0.19 | 0.06 | 0.11 |
| Machines | Independents | 0.17 | 0.02 | -0.02 |
| Republicans | Democrats | 0.34 | 0.05 | -0.01 |
| Democrats | Independents | 0.43 | 0.03 | -0.04 |
| Independents | Republicans | 0.39 | 0.36 | -0.03 |
| Democrats | Democrats$^{Rep}$ | 0.38 | -0.05 | -0.02 |
| Democrats | Democrats$^{Ind}$ | 0.46 | 0.00 | -0.03 |
| Republicans | Republicans$^{Dem}$ | 0.37 | -0.01 | 0.00 |
| Republicans | Republicans$^{Ind}$ | 0.35 | 0.15 | -0.03 |
| Independents | Independents$^{Rep}$ | 0.37 | 0.18 | 0.04 |
| Independents | Independents$^{Dem}$ | 0.43 | -0.04 | -0.04 |
| Republicans$^{Dem}$ | Republicans$^{Ind}$ | 0.49 | 0.01 | -0.04 |
| Democrats$^{Ind}$ | Democrats$^{Rep}$ | 0.44 | 0.57 | 0.10 |
| Independents$^{Rep}$ | Independents$^{Dem}$ | 0.37 | 0.06 | -0.05 |

Table 5: Disagreement across different annotation datasets.

## D.2 Understanding Human Agreement

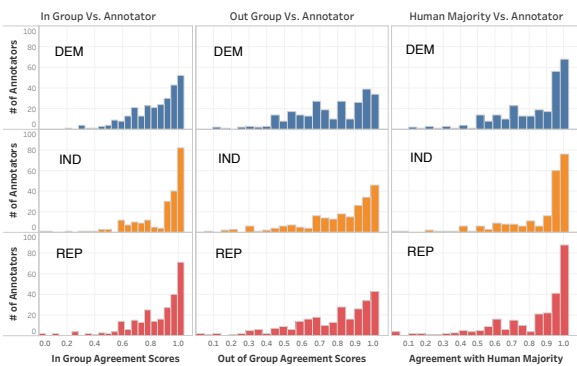

Figure 14: Histograms of the agreement scores for human annotators.(top to bottom) The rows denote political leaning, also represented with colors, Democrat, Independent, and Republican. (left to right) The first set of graphs capture the agreement for annotators against other annotators from the same political leaning. Second set captures the agreement of the annotator against the majority of the opposing groups. Third set represents the agreement against the majority opinion of the entire population of annotators.

Moving beyond Cohen's $k$, in this section we study the agreement of a single annotator against three perspectives. (1) other annotators who are from the same political leaning (in-group), (2) the majority of the other annotators from the opposing groups (out-group), and (3) the overall human annotator pool majority (humans). We study these agreements by calculating the *match score*, i.e., the ratio of matched labels for the perspective considered divided by the number of total data items the annotator was exposed. We use this metric to supplement the analysis from the prior sections. Figure 14 contains histograms for each perspective considered based the political leaning of the annotators.

Based on the analysis from the Figure 14, there are some interesting observations for **this study**. Based on the highest agreement bins from the figure and the highest bin (0.9 to 1.0). The number of **Democrat** leaning annotators that showed the strongest agreement was surpassed by other political leanings across all the three perspectives considered. Majority of the **Independents** showed the strongest agreement across the three perspectives, bypassing their Democrat and Republican leaning annotators. And majority of the **Republican** leaning annotators showed the strongest agreement with the overall human majority label.

Additional results to the discussion on Figure 3. The tables below demonstrate the confusion matrix for each moderator pair.

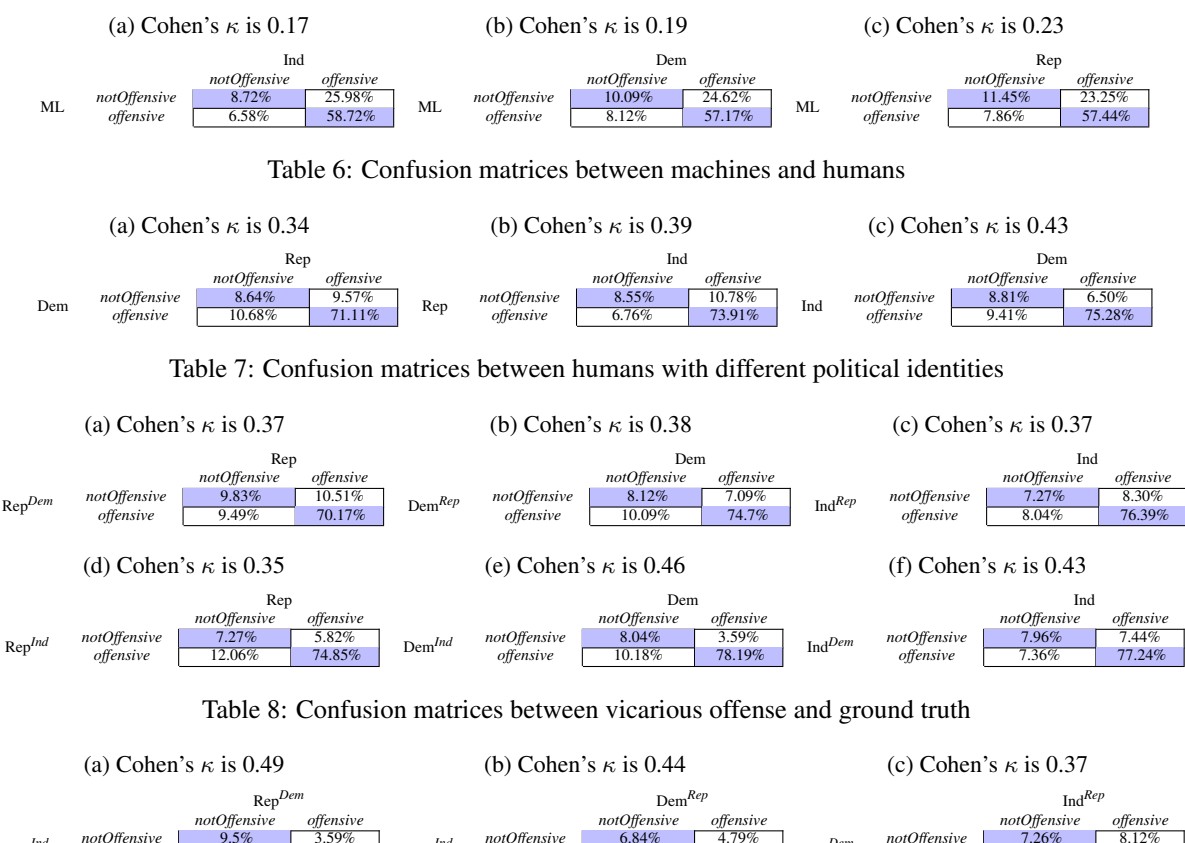

(a) Cohen's $\kappa$ is 0.17

|  | Ind | |
|---|---|---|
|  | notOffensive | offensive |
| ML notOffensive | 8.72% | 25.98% |
| ML offensive | 6.58% | 58.72% |

(b) Cohen's $\kappa$ is 0.19

|  | Dem | |
|---|---|---|
|  | notOffensive | offensive |
| ML notOffensive | 10.09% | 24.62% |
| ML offensive | 8.12% | 57.17% |

(c) Cohen's $\kappa$ is 0.23

|  | Rep | |
|---|---|---|
|  | notOffensive | offensive |
| ML notOffensive | 11.45% | 23.25% |
| ML offensive | 7.86% | 57.44% |

Table 6: Confusion matrices between machines and humans

(a) Cohen's $\kappa$ is 0.34

|  | Rep | |
|---|---|---|
|  | notOffensive | offensive |
| Dem notOffensive | 8.64% | 9.57% |
| Dem offensive | 10.68% | 71.11% |

(b) Cohen's $\kappa$ is 0.39

|  | Ind | |
|---|---|---|
|  | notOffensive | offensive |
| Rep notOffensive | 8.55% | 10.78% |
| Rep offensive | 6.76% | 73.91% |

(c) Cohen's $\kappa$ is 0.43

|  | Dem | |
|---|---|---|
|  | notOffensive | offensive |
| Ind notOffensive | 8.81% | 6.50% |
| Ind offensive | 9.41% | 75.28% |

Table 7: Confusion matrices between humans with different political identities

(a) Cohen's $\kappa$ is 0.37

|  | Rep | |
|---|---|---|
|  | notOffensive | offensive |
| $Rep^{Dem}$ notOffensive | 9.83% | 10.51% |
| $Rep^{Dem}$ offensive | 9.49% | 70.17% |

(b) Cohen's $\kappa$ is 0.38

|  | Dem | |
|---|---|---|
|  | notOffensive | offensive |
| $Dem^{Rep}$ notOffensive | 8.12% | 7.09% |
| $Dem^{Rep}$ offensive | 10.09% | 74.7% |

(c) Cohen's $\kappa$ is 0.37

|  | Ind | |
|---|---|---|
|  | notOffensive | offensive |
| $Ind^{Rep}$ notOffensive | 7.27% | 8.30% |
| $Ind^{Rep}$ offensive | 8.04% | 76.39% |

(d) Cohen's $\kappa$ is 0.35

|  | Rep | |
|---|---|---|
|  | notOffensive | offensive |
| $Rep^{Ind}$ notOffensive | 7.27% | 5.82% |
| $Rep^{Ind}$ offensive | 12.06% | 74.85% |

(e) Cohen's $\kappa$ is 0.46

|  | Dem | |
|---|---|---|
|  | notOffensive | offensive |
| $Dem^{Ind}$ notOffensive | 8.04% | 3.59% |
| $Dem^{Ind}$ offensive | 10.18% | 78.19% |

(f) Cohen's $\kappa$ is 0.43

|  | Ind | |
|---|---|---|
|  | notOffensive | offensive |
| $Ind^{Dem}$ notOffensive | 7.96% | 7.44% |
| $Ind^{Dem}$ offensive | 7.36% | 77.24% |

Table 8: Confusion matrices between vicarious offense and ground truth

(a) Cohen's $\kappa$ is 0.49

|  | $Rep^{Dem}$ | |
|---|---|---|
|  | notOffensive | offensive |
| $Rep^{Ind}$ notOffensive | 9.5% | 3.59% |
| $Rep^{Ind}$ offensive | 10.86% | 76.05% |

(b) Cohen's $\kappa$ is 0.44

|  | $Dem^{Rep}$ | |
|---|---|---|
|  | notOffensive | offensive |
| $Dem^{Ind}$ notOffensive | 6.84% | 4.79% |
| $Dem^{Ind}$ offensive | 8.38% | 79.99% |

(c) Cohen's $\kappa$ is 0.37

|  | $Ind^{Rep}$ | |
|---|---|---|
|  | notOffensive | offensive |
| $Ind^{Dem}$ notOffensive | 7.26% | 8.12% |
| $Ind^{Dem}$ offensive | 8.29% | 76.33% |

Table 9: Contrasting vicarious offense predictions

Figure 15: The confusion matrices below includes results from comparing the human annotation input against other closed systems such as ChatGPT (gpt3.5turbo, v3.5) and Perspective API. In the figures, gpt_label is input from ChatGPT and papi_label is input from Perspective API. For papi_label as per the API documentation[8], we used 0.7 as the threshold for toxicity classification. For obtaining the gpt_label, we use the prompt "You are an emotionally intelligent assistant. Classify the intent of the user's text to be offensive/disrespectful to someone with ONLY ONE OF THE FOLLOWING CHOICES: 1. Very offensive, 2. Extremely offensive, 3. Slightly offensive, 4. Moderately offensive, or 5. Not at all offensive."

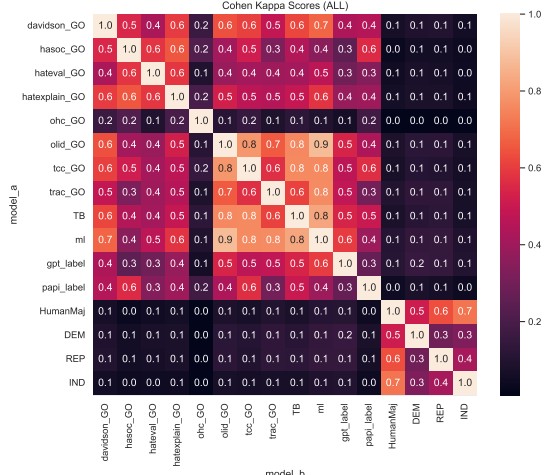

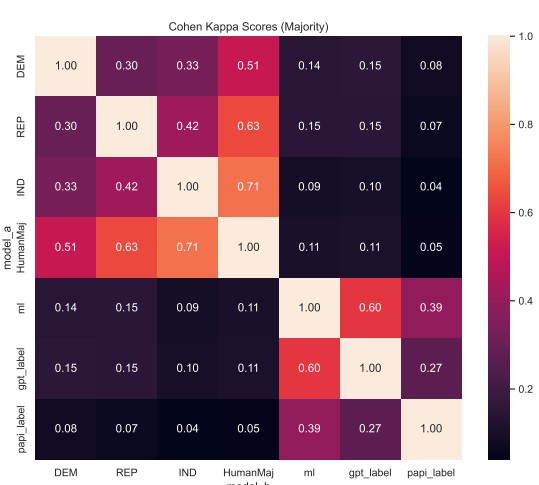

Figure 16: Additional Examples to the Examples in Figure1

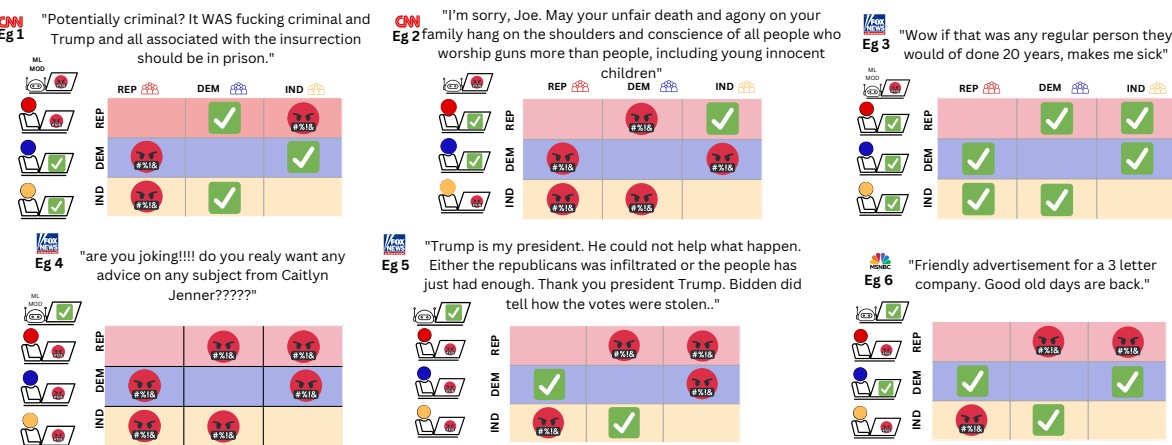

Figure 17: Illustrative examples highlighting nuanced inconsistencies between machine moderators and human moderators with different political leanings. For every comment, majority vote is used to aggregate individual machine and human moderator's verdicts. An angry emoji and a green checkbox indicate *offensive* and *notOffensive* labels, respectively. These real-world examples are drawn from comments on YouTube news videos of three major US cable news networks: Fox News, CNN, and MSNBC. Each example is annotated by 20 human moderators with at least six Republicans, Democrats, and Independents. Nine well-known offensive speech data sets are used to create nine machine moderators. The grid summarizes vicarious offense where annotators belonging to the row political identity are asked to predict vicarious offense perspectives of the two other political identities mentioned in the columns.