# OpenReview forum: "Vicarious Offense and Noise Audit of Offensive Speech Classifiers: Unifying Human and Machine Disagreement on What is Offensive"
_EMNLP/2023/Conference — EMNLP 2023 Main_

### Official Review · Reviewer_G1Gg · 2023-07-28

**Soundness:** 4

**Excitement:**

4: Strong: This paper deepens the understanding of some phenomenon or lowers the barriers to an existing research direction.

**Paper Topic And Main Contributions:**

The work conducted a noise audit of nine offensive speech classifiers on a dataset of more than 92 million YouTube comments, revealing considerable variations in the results. Also, it annotated a dataset with views from people from different political parties regarding their perception of offense and also about vicarious offense, that is, to predict offense for others who do not share the same political belief. Finally, it analyzed human and machine moderators' agreement on what is offensive.

**Reasons To Accept:**

- The release of a novel dataset that can be valuable for further research regarding vicarious offense.
- The annotation process is well explained.
- Experimental evaluation is complete: comparing machine and human moderators, generating informative results, and discussing the impact of what was achieved.

**Reasons To Reject:**

- Since each machine moderator is trained on a different dataset, they are expected to have a low agreement. Training on a concatenation of datasets or evaluating more public APIs, such as Perspective API, would generate more informative noise audit results.
- Some presentation problems make the last two sections difficult to follow. I suggested improvements in the corresponding section from review.

**Reproducibility:**

4: Could mostly reproduce the results, but there may be some variation because of sample variance or minor variations in their interpretation of the protocol or method.

**Reviewer Confidence:**

4: Quite sure. I tried to check the important points carefully. It's unlikely, though conceivable, that I missed something that should affect my ratings.

**Typos Grammar Style And Presentation Improvements:**

- Please cite the Appendix subsections in the paper so we can refer easily to them.
- Table 1 is not cited in the paper.
- Line 381: The cited result (0.43) is extracted from the Appendix. Consider reorganizing so the result appears in the main paper.
- Figure 3 caption is incorrect. A cell [i, j] is not referring to machine moderators' agreement since the figure is also showing human moderators' agreement.
- Line 478: Figure reference is missing.
- Figure 4 uses a different vicarious offense notation from what was previously explained in the paper (in lines 401-404).
- The results using ChatGPT to predict the vicarious offense are in the Discussion and Conclusion section, which should be in the Results section.
- In the Appendix, Figures 5, 6, and 7 are not cited.
- In the Appendix, Section B.1 is empty.

---

> ### Author Rebuttal · Authors · 2023-08-28
>
> We are glad that all three reviewers found our work exciting and methodologically sound. We thank them for their constructive criticism, improvement suggestions, and pointers to relevant literature. We have already addressed most of them in the rebuttal. We will address all of them in the final version.
>
> We appreciate the comments about our work.
>
> >Some presentation problems make the last two sections difficult to follow. I suggested improvements in the corresponding section from review.
>
> We will include all the changes in the final version of the paper and work on the formatting to avoid any misaligned sections.
>
> >Since each machine moderator is trained on a different dataset, they are expected to have a low agreement. Training on a concatenation of datasets or evaluating more public APIs, such as Perspective API, would generate more informative noise audit results.
>
> Adding a machine moderator trained on the union of the datasets is an excellent suggestion. We will include this result in the final version. However, the datasets we use are well-known within the community and there is no documented understanding of how aligned they are on in-the-wild evaluation sets. Our study addresses that gap.
>
> We have included additional results with the Perspective API on the human annotated dataset. In the final version, we will include a noise audit with Perspective API and the model trained on the union of the datasets.
>
> ### Additional Results
>
> These models are from the GeneralOffense models and TB is toxic BERT.
> -   **ml** - this is the majority label from all the models (GO and TB)
> -   **gpt_label** - this is the label when ChatGPT 3.5 is used for classification (prompt is included in details). "*You are an emotionally intelligent assistant. Classify the intent of the user's text to be offensive/disrespectful to someone with ONLY ONE OF THE FOLLOWING CHOICES: 1. Very offensive, 2. Extremely offensive, 3. Slightly offensive, 4. Moderately offensive, or 5. Not at all offensive.*". This prompt is similar to the one we've used for analysis of the vicarious offense.
> - **papi_label** - this label is from Perspective API Scores. Following the guidelines from the [documents](https://developers.perspectiveapi.com/s/about-the-api-score?language=en_US), we used the cut off as 0.7.
> -   **HumanMaj** - this is the majority opinion of the humans.
> -   **DEM**,**REP**, and **IND** - majority opinion of all the Democrat, Republican, and Independent leaning human annotators respectively.
>
> **Cohen's Kappa**
> |   |   DEM |   REP |   IND |   HumanMaj |   ml |   gpt_label |   papi_label |
> |------------|-------|-------|-------|------------|------|-------------|--------------|
> | DEM        |  1.00 |  0.30 |  0.33 |       0.51 | 0.14 |        0.15 |         0.08 |
> | REP        |  0.30 |  1.00 |  0.42 |       0.63 | 0.15 |        0.15 |         0.07 |
> | IND        |  0.33 |  0.42 |  1.00 |       0.71 | 0.09 |        0.10 |         0.04 |
> | HumanMaj   |  0.51 |  0.63 |  0.71 |       1.00 | 0.11 |        0.11 |         0.05 |
> | ml         |  0.14 |  0.15 |  0.09 |       0.11 | 1.00 |        0.60 |         0.39 |
> | gpt_label  |  0.15 |  0.15 |  0.10 |       0.11 | 0.60 |        1.00 |         0.27 |
> | papi_label |  0.08 |  0.07 |  0.04 |       0.05 | 0.39 |        0.27 |         1.00 |
>
> **Recall**
> |  |   DEM |   REP |   IND |   HumanMaj |   ml |   gpt_label |   papi_label |
> |------------|-------|-------|-------|------------|------|-------------|--------------|
> | DEM        |  1.00 |  0.63 |  0.64 |       0.70 | 0.61 |        0.60 |         0.60 |
> | REP        |  0.67 |  1.00 |  0.69 |       0.77 | 0.64 |        0.62 |         0.61 |
> | IND        |  0.72 |  0.74 |  1.00 |       0.83 | 0.61 |        0.60 |         0.58 |
> | HumanMaj   |  0.87 |  0.90 |  0.88 |       1.00 | 0.65 |        0.62 |         0.61 |
> | ml         |  0.56 |  0.56 |  0.54 |       0.54 | 1.00 |        0.78 |         0.74 |
> | gpt_label  |  0.57 |  0.56 |  0.54 |       0.54 | 0.83 |        1.00 |         0.70 |
> | papi_label |  0.56 |  0.55 |  0.53 |       0.54 | 0.75 |        0.68 |         1.00 |
>
> **F1-Scores**
>
> |  |   DEM |   REP |   IND |   HumanMaj |   ml |   gpt_label |   papi_label |
> |------------|-------|-------|-------|------------|------|-------------|--------------|
> | DEM        |  1.00 |  0.65 |  0.66 |       0.75 | 0.55 |        0.57 |         0.40 |
> | REP        |  0.65 |  1.00 |  0.71 |       0.82 | 0.54 |        0.56 |         0.37 |
> | IND        |  0.66 |  0.71 |  1.00 |       0.86 | 0.50 |        0.52 |         0.34 |
> | HumanMaj   |  0.75 |  0.82 |  0.86 |       1.00 | 0.50 |        0.52 |         0.33 |
> | ml         |  0.55 |  0.54 |  0.50 |       0.50 | 1.00 |        0.80 |         0.66 |
> | gpt_label  |  0.57 |  0.56 |  0.52 |       0.52 | 0.80 |        1.00 |         0.57 |
> | papi_label |  0.40 |  0.37 |  0.34 |       0.33 | 0.66 |        0.57 |         1.00 |

---

### Official Review · Reviewer_Vd66 · 2023-08-05

**Soundness:** 4

**Excitement:**

4: Strong: This paper deepens the understanding of some phenomenon or lowers the barriers to an existing research direction.

**Missing References:**

There are two works that perform studies on offense in political discourse:

1. Hate Towards the Political Opponent: A Twitter Corpus Study of the 2020 US Elections on the Basis of Offensive Speech and Stance Detection, Lara Grimminger, Roman Klinger, 2021. (work collects offensive tweets from democrats and republicans targeting the other community)

2. Listening to Affected Communities to Define Extreme Speech: Dataset and Experiments, Antonis Maronikolakis, Axel Wisiorek, Leah Nann, Haris Jabbar, Sahana Udupa, Hinrich Schuetze, 2022. (one of the domains examined is hate speech in political discourse, collecting data targeting politicians, state, civil rights advocates, etc.)

While these two differ from this work since they only collect data for hate speech in political discourse and do not perform a noise audit, I find they are still pertinent (slightly contradicting lines 167-169).

**Paper Topic And Main Contributions:**

This work operates in the domain of political discourse, performing a large-scale study on the perception of offense from across the political spectra as well as from machine moderators. Further, a dataset produced by their study is released, labeled additionally for "vicarious offense".

**Questions For The Authors:**

1. I was wondering whether it would be useful to add an example case of vicarious offense in the introduction to better showcase this concept. Even though the definition is adequate, this is a new concept and would therefore benefit from a readily available example. Maybe you could repurpose the Fig. 1 caption to specifically point out what vicarious offense is.

2. Can this study have a temporal element as well? It would be interesting to see how the findings evolve throughout the years.

3. Is there a reason you did not use the Perspective API? You did use TCC (model 8 in your list), but I am wondering why the Perspective API was avoided. While I am not advocating for its use, a lot of contemporary work uses it so I found this decision interesting. If there is an explicit reason behind this, including it in the paper would be informative.

4. For RQ2, I would have liked to have seen self-alignment as a baseline: what is the alignment within Democrats (Dem^Dem)? How well are Democrats at predicting what other Democrats find offensive? This would be interesting as a comparison.

**Reasons To Accept:**

1. A novel study analyzing perceptions of offense from different political spectra, as well as alignment with machine learning models. This study is vital in current political discourse and discussion on hate speech.

2. Description of crowdsourced annotators is thorough and gives a solid picture of the annotator pool.

3. The list of machine moderators is extensive, leading to a thorough analysis.

4. Findings are interesting and the experiments conducted robustly.

**Reasons To Reject:**

Nothing significant. See questions for comments/suggestions for improvement.

**Reproducibility:**

3: Could reproduce the results with some difficulty. The settings of parameters are underspecified or subjectively determined; the training/evaluation data are not widely available.

**Reviewer Confidence:**

4: Quite sure. I tried to check the important points carefully. It's unlikely, though conceivable, that I missed something that should affect my ratings.

**Typos Grammar Style And Presentation Improvements:**

lines 193 and 210 and 232: Usually footnotes are added after punctuation for more compact text. Such as this: text.1

---

> ### Author Rebuttal · Authors · 2023-08-28
>
> We are glad that all three reviewers found our work exciting and methodologically sound. We thank them for their constructive criticism, improvement suggestions, and pointers to relevant literature. We have already addressed most of them in the rebuttal. We will address all of them in the final version.
>
> We thank the reviewer for their feedback on our work. We will incorporate the suggestions and references to the final version of the paper. As for the questions raised;
>
> >Q1: I was wondering whether it would be useful to add an example case of vicarious offense in the introduction to better showcase this concept. Even though the definition is adequate, this is a new concept and would therefore benefit from a readily available example. Maybe you could repurpose the Fig. 1 caption to specifically point out what vicarious offense is.
>
> Question 1, on adding the vicarious offense to the image caption. We will rephrase the caption to include details about the vicarious offense for the readers.
>
> >Q2: Can this study have a temporal element as well? It would be interesting to see how the findings evolve throughout the years.
>
> Question 2, we analyzed the overall toxicity of the dataset in a temporal dimension (appendix Figure 11). But due to space constraints, we had to move the analysis to the Appendix.
>
> >Q3: Is there a reason you did not use the Perspective API? You did use TCC (model 8 in your list), but I am wondering why the Perspective API was avoided. While I am not advocating for its use, a lot of contemporary work uses it so I found this decision interesting. If there is an explicit reason behind this, including it in the paper would be informative.
>
> We did analyze with the ChatGPT and the Perspective API on the overall alignment of the offensiveness against the current set of models and human annotators (see the table). Looking at the results (Cohen’s Kappa, Figures 1A and 1B), the humans agree more with the ML and ChatGPT label and less with the Perspective API. This is an observation that is common looking at F1 and recall (Figures 2 and Figure 3). We will include these findings in the final version of the paper.
>
> These models are from the GeneralOffense models and TB is toxic BERT.
> -   **ml** - this is the majority label from all the models (GO and TB)
> -   **gpt_label** - this is the label when ChatGPT 3.5 is used for classification (prompt is included in details). "*You are an emotionally intelligent assistant. Classify the intent of the user's text to be offensive/disrespectful to someone with ONLY ONE OF THE FOLLOWING CHOICES: 1. Very offensive, 2. Extremely offensive, 3. Slightly offensive, 4. Moderately offensive, or 5. Not at all offensive.*". This prompt is similar to the one we've used for analysis of the vicarious offense.
> - **papi_label** - this label is from Perspective API Scores. Following the guidelines from the [documents](https://developers.perspectiveapi.com/s/about-the-api-score?language=en_US), we used the cut off as 0.7.
> -   **HumanMaj** - this is the majority opinion of the humans.
> -   **DEM**,**REP**, and **IND** - majority opinion of all the Democrat, Republican, and Independent leaning human annotators respectively.
>
> **Cohen's Kappa**
> |   |   DEM |   REP |   IND |   HumanMaj |   ml |   gpt_label |   papi_label |
> |------------|-------|-------|-------|------------|------|-------------|--------------|
> | DEM        |  1.00 |  0.30 |  0.33 |       0.51 | 0.14 |        0.15 |         0.08 |
> | REP        |  0.30 |  1.00 |  0.42 |       0.63 | 0.15 |        0.15 |         0.07 |
> | IND        |  0.33 |  0.42 |  1.00 |       0.71 | 0.09 |        0.10 |         0.04 |
> | HumanMaj   |  0.51 |  0.63 |  0.71 |       1.00 | 0.11 |        0.11 |         0.05 |
> | ml         |  0.14 |  0.15 |  0.09 |       0.11 | 1.00 |        0.60 |         0.39 |
> | gpt_label  |  0.15 |  0.15 |  0.10 |       0.11 | 0.60 |        1.00 |         0.27 |
> | papi_label |  0.08 |  0.07 |  0.04 |       0.05 | 0.39 |        0.27 |         1.00 |
>
> **Recall**
> |  |   DEM |   REP |   IND |   HumanMaj |   ml |   gpt_label |   papi_label |
> |------------|-------|-------|-------|------------|------|-------------|--------------|
> | DEM        |  1.00 |  0.63 |  0.64 |       0.70 | 0.61 |        0.60 |         0.60 |
> | REP        |  0.67 |  1.00 |  0.69 |       0.77 | 0.64 |        0.62 |         0.61 |
> | IND        |  0.72 |  0.74 |  1.00 |       0.83 | 0.61 |        0.60 |         0.58 |
> | HumanMaj   |  0.87 |  0.90 |  0.88 |       1.00 | 0.65 |        0.62 |         0.61 |
> | ml         |  0.56 |  0.56 |  0.54 |       0.54 | 1.00 |        0.78 |         0.74 |
> | gpt_label  |  0.57 |  0.56 |  0.54 |       0.54 | 0.83 |        1.00 |         0.70 |
> | papi_label |  0.56 |  0.55 |  0.53 |       0.54 | 0.75 |        0.68 |         1.00 |
>
> **F1-Scores**
>
> |  |   DEM |   REP |   IND |   HumanMaj |   ml |   gpt_label |   papi_label |
> |------------|-------|-------|-------|------------|------|-------------|--------------|
> | DEM        |  1.00 |  0.65 |  0.66 |       0.75 | 0.55 |        0.57 |         0.40 |
> | REP        |  0.65 |  1.00 |  0.71 |       0.82 | 0.54 |        0.56 |         0.37 |
> | IND        |  0.66 |  0.71 |  1.00 |       0.86 | 0.50 |        0.52 |         0.34 |
> | HumanMaj   |  0.75 |  0.82 |  0.86 |       1.00 | 0.50 |        0.52 |         0.33 |
> | ml         |  0.55 |  0.54 |  0.50 |       0.50 | 1.00 |        0.80 |         0.66 |
> | gpt_label  |  0.57 |  0.56 |  0.52 |       0.52 | 0.80 |        1.00 |         0.57 |
> | papi_label |  0.40 |  0.37 |  0.34 |       0.33 | 0.66 |        0.57 |         1.00 |
>
> >Q4: For RQ2, I would have liked to have seen self-alignment as a baseline: what is the alignment within Democrats (Dem^Dem)? How well are Democrats at predicting what other Democrats find offensive? This would be interesting as a comparison.
>
> Question 4, it is an interesting question. We did not ask this in our study. But we will include it as a future consideration in the final version of the paper.

---

### Official Review · Reviewer_Gr8n · 2023-08-11

**Soundness:** 3

**Excitement:**

4: Strong: This paper deepens the understanding of some phenomenon or lowers the barriers to an existing research direction.

**Missing References:**

None in my opinion.

**Paper Topic And Main Contributions:**

The paper focuses on first-person offense and vicarious offense in US-based political conversations. The paper then investigates 3 research questions based on the alignment between machine-machine, human-human, and machine-human moderators to identify first-person and vicarious offense present in the conversations.

**Questions For The Authors:**

A. What is the motivation to use the generic datasets for the special case of identifying offenses in US-based political conversations? It is possible models may not learn well because the datasets do not provide enough information related to democrats, republicans, and independent groups based offense.

B. Do the authors try to use few-shot ChatGPT/Perspective API or other open-source APIs for vicarious and first-person offense identification? It is possible the results might be changed greatly. However, it is still a possibility that can be understood after running the experiments.

**Reasons To Accept:**

The idea is interesting and exciting.
The need to focus on vicarious offense is indeed important from the societal point of view.
Results and discussions are properly backed by political theories.
The comparison of human-human moderators is well discussed and presented.
Overall, the writing of the paper is impressive as it is well written and understood easily.

**Reasons To Reject:**

I am not convinced how authors have claimed about machine moderators. I feel that not enough importance is given to the preparation of machine moderators. Similar architectures of BERT /RoBERTa models are trained on different datasets and used as machine moderators to evaluate their performance on unseen political data. Due to the very simple architecture and the fact that I think these datasets are general datasets (with high imbalance) that are not entirely based on political data, it is possible that these models do not perform well on first-person offense detection and therefore have not been well aligned. I like that the authors in the discussion section used ChatGPT for identifying vicarious offense, but they have not shown how ChatGPT performs in identifying first-person offense compared to the models used in the study. Nevertheless, major claims are made about the inability of machine moderators to identify offense. In my opinion, these claims need to be backed up by better models, APIs such as Perspective API, or recent open-source/APIs-based large language models.

**Reproducibility:**

4: Could mostly reproduce the results, but there may be some variation because of sample variance or minor variations in their interpretation of the protocol or method.

**Reviewer Confidence:**

4: Quite sure. I tried to check the important points carefully. It's unlikely, though conceivable, that I missed something that should affect my ratings.

**Typos Grammar Style And Presentation Improvements:**

I think that the illustrations and tables should be better placed. The figures that are mentioned on one page are located after 2 pages, which sometimes makes it difficult to follow them. It is always better to place the tables and figures before you mention them.

Typos: Figure ?? in section 5.3
A.4, B.1 only headings are mentioned

---

> ### Author Rebuttal · Authors · 2023-08-28
>
> We are glad that all three reviewers found our work exciting and methodologically sound. We thank them for their constructive criticism, improvement suggestions, and pointers to relevant literature. We have already addressed most of them in the rebuttal. We will address all of them in the final version.
>
> We thank the reviewer for their feedback on our own. We will address the improvements suggested to the paper along with sections to clarify the issues raised in the reviews.
>
> > A. What is the motivation to use the generic datasets for the special case of identifying offenses in US-based political conversations? It is possible models may not learn well because the datasets do not provide enough information related to democrats, republicans, and independent groups based offense.
>
> Question A: As the reviewer suggests, we have included ChatGPT and Perspective API results. A key takeaway from our new results on first-person offense with ChatGPT and Perspective API is that they are *not* significantly better aligned with humans as compared to other machine moderators.
>
> We agree with the reviewer that generic datasets may not fully capture the nuances of the current US political schism. However, several of these datasets are sourced from Twitter, Reddit, and GAB and are likely to contain political content. For example, in OLID [Zampieri et al. https://aclanthology.org/N19-1144/], authors sampled replies from both sides of the political spectrum, using politically charged keywords such as MAGA, Antifa, Conservatives, Liberals, etc. Also, our research question here is how often models trained on these datasets agree when evaluated in the wild. The nine datasets that we selected are well-known within the community and the audit reveals that their in-the-wild agreement is low. Our ablation analysis indicates qualitatively similar results in other datasets unrelated to US politics as well (Line 1082 reports results on BBC News comments).
> We are happy to include any additional methods the reviewer recommends (e.g., PaLM 2). Also, we need to keep in mind that running experiments on millions of data points is computationally prohibitive (PaLM2 allows 30 requests per minute, and ChatGPT costs will be prohibitively large). The analysis of the human annotated dataset of 2,310 items with Perspective API and ChatGPT took ~1 hr working around with the API restrictions in place.
>
> Finally, as the API versions can be modified externally (e.g., a recent study reports ChatGPT’s declining math skills [Chen, Zaharia, Zhou et al., 2023, https://arxiv.org/pdf/2307.09009.pdf], in can have unforeseen reproducibility challenges. In contrast, the offline models (will be made publicly available upon acceptance) we use enable other researchers to reproduce our work.
>
> >B. Do the authors try to use few-shot ChatGPT/Perspective API or other open-source APIs for vicarious and first-person offense identification? It is possible the results might be changed greatly. However, it is still a possibility that can be understood after running the experiments.
>
> In response to Question B, about utilizing the few-shot approaches ChatGPT/Perspective API: We did analyze with the ChatGPT and the Perspective API on the overall alignment of the offensiveness against the current set of models and human annotators (see the table). Looking at the results (Cohen’s Kappa, Figures 1A and 1B), the humans agree more with the ML and ChatGPT label and less with the Perspective API. This is an observation that is common looking at F1 and recall (Figures 2 and Figure 3). We will include these findings in the final version of the paper.
>
> ### Additional Results
>
> These models are from the GeneralOffense models and TB is toxic BERT.
> -   **ml** - this is the majority label from all the models (GO and TB)
> -   **gpt_label** - this is the label when ChatGPT 3.5 is used for classification (prompt is included in details). "*You are an emotionally intelligent assistant. Classify the intent of the user's text to be offensive/disrespectful to someone with ONLY ONE OF THE FOLLOWING CHOICES: 1. Very offensive, 2. Extremely offensive, 3. Slightly offensive, 4. Moderately offensive, or 5. Not at all offensive.*". This prompt is similar to the one we've used for analysis of the vicarious offense.
> - **papi_label** - this label is from Perspective API Scores. Following the guidelines from the [documents](https://developers.perspectiveapi.com/s/about-the-api-score?language=en_US), we used the cut off as 0.7.
> -   **HumanMaj** - this is the majority opinion of the humans.
> -   **DEM**,**REP**, and **IND** - majority opinion of all the Democrat, Republican, and Independent leaning human annotators respectively.
>
> **Cohen's Kappa**
> |   |   DEM |   REP |   IND |   HumanMaj |   ml |   gpt_label |   papi_label |
> |------------|-------|-------|-------|------------|------|-------------|--------------|
> | DEM        |  1.00 |  0.30 |  0.33 |       0.51 | 0.14 |        0.15 |         0.08 |
> | REP        |  0.30 |  1.00 |  0.42 |       0.63 | 0.15 |        0.15 |         0.07 |
> | IND        |  0.33 |  0.42 |  1.00 |       0.71 | 0.09 |        0.10 |         0.04 |
> | HumanMaj   |  0.51 |  0.63 |  0.71 |       1.00 | 0.11 |        0.11 |         0.05 |
> | ml         |  0.14 |  0.15 |  0.09 |       0.11 | 1.00 |        0.60 |         0.39 |
> | gpt_label  |  0.15 |  0.15 |  0.10 |       0.11 | 0.60 |        1.00 |         0.27 |
> | papi_label |  0.08 |  0.07 |  0.04 |       0.05 | 0.39 |        0.27 |         1.00 |
>
> **Recall**
> |  |   DEM |   REP |   IND |   HumanMaj |   ml |   gpt_label |   papi_label |
> |------------|-------|-------|-------|------------|------|-------------|--------------|
> | DEM        |  1.00 |  0.63 |  0.64 |       0.70 | 0.61 |        0.60 |         0.60 |
> | REP        |  0.67 |  1.00 |  0.69 |       0.77 | 0.64 |        0.62 |         0.61 |
> | IND        |  0.72 |  0.74 |  1.00 |       0.83 | 0.61 |        0.60 |         0.58 |
> | HumanMaj   |  0.87 |  0.90 |  0.88 |       1.00 | 0.65 |        0.62 |         0.61 |
> | ml         |  0.56 |  0.56 |  0.54 |       0.54 | 1.00 |        0.78 |         0.74 |
> | gpt_label  |  0.57 |  0.56 |  0.54 |       0.54 | 0.83 |        1.00 |         0.70 |
> | papi_label |  0.56 |  0.55 |  0.53 |       0.54 | 0.75 |        0.68 |         1.00 |
>
> **F1-Scores**
>
> |  |   DEM |   REP |   IND |   HumanMaj |   ml |   gpt_label |   papi_label |
> |------------|-------|-------|-------|------------|------|-------------|--------------|
> | DEM        |  1.00 |  0.65 |  0.66 |       0.75 | 0.55 |        0.57 |         0.40 |
> | REP        |  0.65 |  1.00 |  0.71 |       0.82 | 0.54 |        0.56 |         0.37 |
> | IND        |  0.66 |  0.71 |  1.00 |       0.86 | 0.50 |        0.52 |         0.34 |
> | HumanMaj   |  0.75 |  0.82 |  0.86 |       1.00 | 0.50 |        0.52 |         0.33 |
> | ml         |  0.55 |  0.54 |  0.50 |       0.50 | 1.00 |        0.80 |         0.66 |
> | gpt_label  |  0.57 |  0.56 |  0.52 |       0.52 | 0.80 |        1.00 |         0.57 |
> | papi_label |  0.40 |  0.37 |  0.34 |       0.33 | 0.66 |        0.57 |         1.00 |

---

### Meta-Review · Area_Chair_YDZV · 2023-09-14

**Recommendation:** 4

**Metareview:**

The reviewers all thought that this paper covered an important topic from an interesting perspective, in a technically sound way. They also raised various questions and concerns throughout their reviews that would be worth addressing in a revised version of the paper.

The main suggestion I want to highlight, that was shared across all reviews, was the choice of machine moderators. Reviewers noted some obvious omissions (e.g., Perspective), and suggested some natural explanations for why they might disagree (divergence in training data), that the authors do not appear to have mentioned. The sense I got from reviews was that this could be addressed via writing edits:
* why were moderators like Perspective not highlighted? (the authors could choose to include additional results from their responses, or they could explicitly provide a reason, e.g., black-box nature of that particular algorithm)
* what are features of the machine moderators, that might contribute to noise? here the authors could go beyond simply listing moderators in a paragraph of text; a table could help to explicitly contrast algorithm/training data differences.

---

### Decision · Program_Chairs · 2023-10-07

**Decision:**

Accept-Main

**Comment:**

The reviewers all thought that this paper covered an important topic from an interesting perspective, in a technically sound way. They also raised various questions and concerns throughout their reviews that would be worth addressing in a revised version of the paper.

The main suggestion I want to highlight, that was shared across all reviews, was the choice of machine moderators. Reviewers noted some obvious omissions (e.g., Perspective), and suggested some natural explanations for why they might disagree (divergence in training data), that the authors do not appear to have mentioned. The sense I got from reviews was that this could be addressed via writing edits:
* why were moderators like Perspective not highlighted? (the authors could choose to include additional results from their responses, or they could explicitly provide a reason, e.g., black-box nature of that particular algorithm)
* what are features of the machine moderators, that might contribute to noise? here the authors could go beyond simply listing moderators in a paragraph of text; a table could help to explicitly contrast algorithm/training data differences.